

# Six simple questions to detect malnutrition or malnutrition risk in elderly women

Tranquilina Gutiérrez-Gómez[1], Ernesto Cortés[2],
Antonio Palazón-Bru[3,4], Isabel Peñarrieta-de Córdova[1],
Vicente Francisco Gil-Guillén[3,4] and Rosa María Ferrer-Diego[5]

[1] Tampico School of Nursing, Autonomous University of Tamaulipas, Tampico, TAMPS, Mexico
[2] Department of Pharmacology, Pediatrics and Organic Chemistry, Miguel Hernández University, San Juan de Alicante, Alicante, Spain
[3] Department of Clinical Medicine, Miguel Hernández University, San Juan de Alicante, Alicante, Spain
[4] Research Unit, Elda Hospital, Elda, Alicante, Spain
[5] Department of Nursing, University of Alicante, San Vicente del Raspeig, Alicante, Spain

Corresponding author
Antonio Palazón-Bru,
antonio.pb23@gmail.com

## ABSTRACT

Of the numerous instruments available to detect nutritional risk, the most widely used is the Mini Nutritional Assessment (MNA), but it takes 15–20 min to complete and its systematic administration in primary care units is not feasible in practice. We developed a tool to evaluate malnutrition risk that can be completed more rapidly using just clinical variables. Between 2008 and 2013, we conducted a cross-sectional study of 418 women aged ≥60 years from Mexico. Our outcome was positive MNA and our secondary variables included were: physical activity, diabetes mellitus, hypertension, educational level, dentition, psychological problems, living arrangements, history of falls, age and the number of tablets taken daily. The sample was divided randomly into two groups: construction and validation. Construction: a risk table was constructed to estimate the likelihood of the outcome, and risk groups were formed. Validation: the area under the ROC curve (AUC) was calculated and we compared the expected and the observed outcomes. The following risk factors were identified: physical activity, hypertension, diabetes, dentition, psychological problems and living with the family. The AUC was 0.77 (95% CI [0.68–0.86], $p < 0.001$). No differences were found between the expected and the observed outcomes ($p = 0.902$). This study presents a new malnutrition screening test for use in elderly women. The test is based on six very simple, quick and easy-to-evaluate questions, enabling the MNA to be reserved for confirmation. However, it should be used with caution until validation studies have been performed in other geographical areas.

## INTRODUCTION

Life expectancy has increased greatly over the past century (*Ferreira et al., 2010*), and the worldwide population of individuals aged over 60 years is expected to triple between

2000 and 2050. This increase reflects an improved health status in younger persons but it also leads to higher rates of chronic degenerative diseases and disability in the elderly (*Shama et al., 2008*).

Active aging requires a good health status together with a good nutritional status, both of which are necessary to maintain a low disease risk and good physical and mental health (*Brigeiro, 2005*). Overall health depends on disease prevention and the promotion of activities associated with healthy habits (*Minkler & Fadem, 2002*; *Celestino, Salazar & Novelo, 2008*).

Older adults are prone to nutritional deficiencies, especially in situations of stress or disease (*Scheidt, Humpherys & Yorgason, 1999*). Malnutrition can arise from various physiological and social factors that lead to adverse consequences, preventing proper body function and impairing the performance of everyday activities (*Brownie, 2006*). Malnutrition, when not treated, is costly both for the individual and for society (*Visvanathan, 2009*).

Most adults over the age of 60 are women (*WHO, 2005*), with different economic, social, political, and cultural factors influencing their aging. The earlier in life a person adopts a healthy lifestyle to prevent morbidity and mortality, the greater the benefits. For example, physical activity should be encouraged from an early age, and all barriers that prevent girls and young women from engaging in physical activity should be eliminated. This will then help elder women maintain their mobility and cope with their everyday activities (*WHO, 1998*).

Several instruments are available to screen for and detect nutritional risk. The most widely used is the Mini Nutritional Assessment (MNA) (*Guigoz & Vellas, 1999*), specifically the MNA short-form version (MNA-SF) (*Rubenstein et al., 2001*). This shortened version correlates strongly with the original version, but only uses the first 6 questions. If the MNA-SF score is $\geq 12$, the results are considered to be normal and the rest of the survey is not administered (sensitivity, 97.9%; specificity, 100%) (*Rubenstein et al., 2001*). However, a MNA-SF score $< 12$ indicates the need to complete the full version of the MNA. The full and short versions require 15–20 and 5–10 min, respectively, to complete.

Currently, no highly reliable instrument exists that can evaluate malnutrition risk within one minute based on just clinical variables. Accordingly, we aimed to develop and validate a new instrument with these characteristics for use in Mexico. This instrument could be used as a pre-screening test to evaluate malnutrition risk in older women who visit primary care units, thereby reducing the waiting times in these units resulting, in part, from application of the currently available instruments. Thus, this instrument could favor the project suggested by the Pan American Health Organization (*PAHO, 2008*).

## MATERIALS & METHODS

### Population, study design, participants and ethical issues

The study population comprised 418 women aged $\geq 60$ years from Tampico, a city in the state of Tamaulipas (Mexico). This state has one of the highest percentages of individuals over 60 years of age (10.3%).

This cross-sectional observational study enrolled women aged ≥60 years, divided into two groups. Group 1 consisted of women who exercised regularly (at least three times per week between August and November 2008) and were members of the "golden age" club at the multidisciplinary gym of the Tampico-Madero University center, of the Autonomous University of Tamaulipas. Group 2 consisted of women who did not exercise regularly between January 15 and February 15, 2013 and who attended the family medicine clinic of the Institute for Social Security and Services for State Workers (ISSSTE). Finally, all the women were required to be literate.

The Autonomous University of Tamaulipas and the ISSSTE approved the study (code: 08-06-01), and all the women provided written informed consent. The study complied with the provisions of the Mexican General Health Law.

## Variables and measurements

The primary study outcome was the risk of malnutrition or malnutrition itself. The women were considered to have a satisfactory nutritional status if they obtained a score ≥12 in the first 6 MNA screening questions. The women who scored fewer than 12 points in these first 6 questions continued with the rest of the questionnaire (full MNA). A final score >23.5 was considered to represent a satisfactory nutritional status. Scores between 17 and 23.5 indicated a risk of malnutrition, and scores <17 indicated malnutrition (*Guigoz, 2006*).

Information about the following variables was collected during a personal interview: physical activity, personal history of diabetes mellitus or hypertension, educational level (secondary and university education, primary or no education), dentition (complete; missing teeth; denture), psychological problems (defined by the presence or absence of dementia or severe or moderate depression) (*Guigoz & Vellas, 1999*), living arrangements (alone; with partner; with family), history of falls over the past year, age, and the number of tablets taken daily.

## Sample size and statistical methods

Construction sample: this included 322 women. In order to contrast an odds ratio (OR) different from 1, the contrast power was calculated from the selected sample, using psychological problems as a factor. The following parameters were used: risk in exposed persons, 0.90; risk in unexposed persons, 0.45; number of exposed persons, 103; number of unexposed persons, 219; type I error, 5%. This all gave a power of 83.40% (*Chow, Wang & Shao, 2008*).

Validation sample: this included 96 women, of whom 55 were either at risk of malnutrition or in a state of malnutrition. With this sample size, setting the confidence level at 95% and expecting an area under the ROC curve (AUC) of 0.75 to detect differences from an AUC of 0.5, we obtained a power of nearly 100% (98.66%) (*Hanley & McNeil, 1982*).

Qualitative variables were described by their frequencies (absolute and relative), whereas means and standard deviations were used to describe quantitative variables. The patients were randomly assigned to one of the samples (construction, 80%; validation, 20%) by generating random numbers. To verify that the two samples were similar,

we performed the Student $t$ test or the $X^2$ test (depending on the type of variable). A confidence interval (CI) was calculated for each parameter using an $\alpha = 5\%$. All the statistical analyses were performed using the IBM SPSS Statistics 19 software.

Construction: a multivariate logistic regression model was constructed to identify the variables associated with malnutrition risk or malnutrition and the adjusted OR were obtained through this model. To determine the combination of variables that could best predict our main outcome, an explanatory variable was introduced into the model for every 25 outcomes (subjects were either at risk of malnutrition or in a state of malnutrition) (a maximum of 7 variables in the model because we had 194 older women at risk of malnutrition or in a state of malnutrition itself: $194 \div 25 = 7.76$) and all the possible combinations that fulfilled this requirement were tested (all possible combinations of 1–7 variables, giving a total of 9,907). For example, physical activity, diabetes, hypertension and higher educational level is a suitable combination with 4 variables ($4 \leq 7$). The combination selected was that with the greatest AUC. The likelihood ratio test was used to measure the goodness-of-fit of the model. Using the $\beta$ coefficients of the multivariate model, a risk table based on the sum of the points was constructed to estimate the probability of malnutrition risk or malnutrition using the Framingham Heart Study methodology. Briefly, this method assigns a score for each category in all the predictive factors, weighting the $\beta$ coefficients of the model. In other words, the method replaces the logit function by points to make predictions of the outcome. Obviously we increase the error in the predictions, but clinicians can use this scoring system in their daily clinical practice (*Sullivan, Massaro & D'Agostino, 2004*). Once the points and their associated risks had been calculated, the AUC to predict malnutrition risk and malnutrition was calculated and four risk groups were defined (based on the quartiles of the points distribution: low, <P25; medium, P25–P50; high, P50–P75; very high, >P75) (*Ramírez-Prado et al., 2015*).

Validation: we calculated the AUC of the scoring system and compared the expected and the observed outcomes in each risk group using the $X^2$ test. The observed outcomes are the real number of people in the sample who are either at risk of malnutrition or in a state of malnutrition, and the expected outcomes are calculated using the probabilities given by the scoring system. For example, if we have 10 patients (2 with the outcome) with $X$ points and the probability of outcome is 25% for this score: observed, 2; expected, 4 (twenty five percent of ten). This is the only way to compare the gold standard with our test, because our test does not take two vales (positive and negative), as it gives us a range of probabilities for each risk group; therefore we cannot calculate sensitivity, specificity, positive and negative predictive values, or positive and negative likelihood ratios.

## RESULTS

Table 1 shows the descriptive characteristics for the construction ($n = 322$) and the validation ($n = 96$) samples. We verified that there were no differences between the two samples ($p : 0.055–0.631$). The outcome ranged between 57.3% and 60.2%. Furthermore, 210 of the 418 women over the age of 60 (157 in the construction sample (48.8%) and 53 in the validation sample (55.2%)) performed regular physical activity. There was a high

**Table 1 Descriptive characteristics and analyses of 418 women over the age of 60 years from Tampico (Tamaulipas, Mexico).** Data from 2013.

| Variable | Construction sample $n = 322$ n(%)/$x \pm s$ | Validation sample $n = 96$ n(%)/$x \pm s$ | p-value | Adj. OR for malnutrition risk or malnutrition (95% CI) | p-value |
|---|---|---|---|---|---|
| Malnutrition risk or malnutrition | 194(60.2) | 55(57.3) | 0.604 | N/A | N/A |
| Physical activity | 157(48.8) | 53(55.2) | 0.267 | 0.32(0.17–0.62) | 0.001 |
| Diabetes | 75(23.3) | 20(20.8) | 0.614 | 1.71(0.87–3.36) | 0.120 |
| Hypertension | 96(29.8) | 33(34.4) | 0.396 | 1.98 (1.09–3.61) | 0.026 |
| Higher educational level | 190(59.0) | 54(56.3) | 0.631 | N/M | N/M |
| Dentition: | | | | | |
| Complete | 44(13.7) | 17(17.7) | 0.319 | 1 | |
| Missing teeth | 126(39.1) | 30(31.3) | | 2.02(0.88–4.66) | 0.100 |
| Denture | 152(47.2) | 49(51.0) | | 2.49(1.06–5.82) | 0.035 |
| Psychological problems | 103(32.0) | 36(37.5) | 0.314 | 9.30(4.35–19.86) | <0.001 |
| Living arrangements: | | | | | |
| Alone | 49(15.2) | 12(12.5) | 0.628 | 1 | |
| With partner | 115(35.7) | 39(40.6) | | 1 | |
| With family | 158(49.1) | 45(46.9) | | 0.67(0.39–1.15) | 0.146 |
| Falls during the last year | 65(20.2) | 26(27.1) | 0.151 | N/M | N/M |
| Age (years) | 66.9 ± 4.8 | 67.3 ± 5.4 | 0.540 | N/M | N/M |
| Number of medications taken daily | 2.5 ± 2.7 | 2.0 ± 2.3 | 0.055 | N/M | N/M |

**Notes.**

$n$(%), absolute frequency (relative frequency); $x \pm s$, mean ± standard deviation; Adj. OR, adjusted odds ratio; CI, confidence interval; N/A, not applicable; N/M, not in the model.

In the model, Dentition and Living arrangements were transformed into two dummy variables. Goodness-of-fit of the model: $X^2 = 97.01$, $p < 0.001$, area under the ROC curve = 0.804.

prevalence of diabetes (20.8–23.3%), hypertension (29.8–34.4%), psychological problems (32.0–37.5%) and falls during the last year (20.2–27.1%). Moreover, only 14.6% (13.7%, construction; 17.7%, validation) of the participants had their full set of teeth.

Construction: the following factors were associated with the risk of malnutrition or malnutrition itself (Table 1): physical activity, dentition, having psychological problems, hypertension, diabetes and living with family. Figure 1 shows the scoring system that was developed using the multivariate model.

Validation: Fig. 2 shows that the AUC was 0.77 (95% CI [0.68–0.86], $p < 0.001$). Figure 3 shows the comparison between the expected and the observed outcomes ($p = 0.902$).

## DISCUSSION

### Summary

We have designed and validated a basic 6-item questionnaire to detect malnutrition or the risk of malnutrition in elderly women. The items (clinical variables) are very easy to administer and proved highly reliable and simple to assess.

### Strengths and limitations

One limitation of this study concerns the possible bias associated with the study sample, as we selected women from a defined area. Thus, this test still requires validation in other

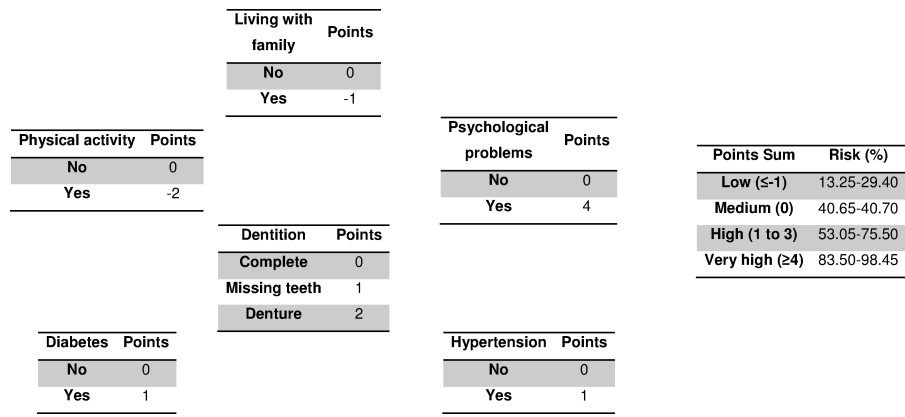

**Figure 1 Prediction scoring system to evaluate malnutrition or the risk of malnutrition in women aged over 60 years in Tampico (Tamaulipas, Mexico).** Data from 2013.

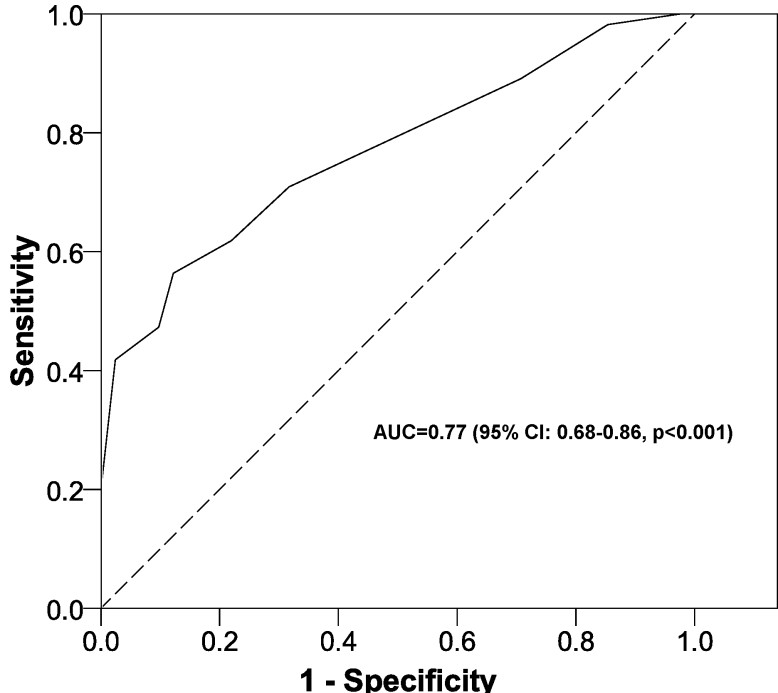

**Figure 2 ROC curve for the scoring system to predict malnutrition or the risk of malnutrition in women aged over 60 years in Tampico (Tamaulipas, México).** Data from 2013. ROC, receiver operating characteristic; AUC, area under the ROC curve; CI, confidence interval.

populations and mainly in the target population, as well as in older men and specific groups ((non-) diabetic or hypertensive patients, taking into account the socioeconomic status and the physical activity status, etc). On the other hand, we have not used nutritional variables to construct our model, because they are included in the MNA test (*Guigoz, 2006*) (outcome of the logistic regression model). However, we have to take into account that
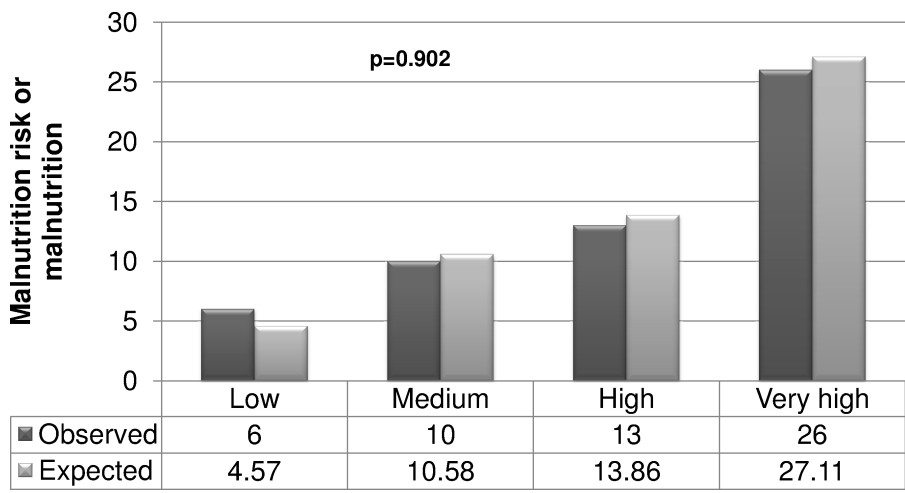

**Figure 3** **Expected and observed outcomes (malnutrition or the risk of malnutrition) in women aged over 60 years in Tampico (Tamaulipas, México).** Data from 2013.

the resulting AUC was 0.77 and no differences were found between expected and observed outcomes ($p = 0.902$).

The strengths of this study include the high discrimination power of the constructed scale, as shown by the area under the ROC curve (approximately 0.8). In addition, this test is easy to give to large populations in a relatively short time, thus increasing its applicability and feasibility. Finally our scoring system has been validated in our population.

## Comparison with existing literature

Nutritional evaluations using the MNA-SF are widely accepted and frequently used for malnutrition screening in elderly populations (*Rubenstein et al., 2001*; *Kaiser et al., 2010*; *Montejano et al., 2013*). However, both the short 6-item version, which includes evaluations of BMI, and the complete 18-item version, which includes two new anthropometric measurements (arm and calf circumference), require some time to complete, placing considerable strain on health centers. The MNA-SF requires the healthcare professional to weigh and measure the patient, even when the patient is immobilized, with the resulting problems concerning undressing or helping a relative who has had an amputation or is handicapped in some way, which also complicates the calculation of the BMI (real or approximate using the relevant formulas), which is required in order to apply the MNA-SF. Consequently, the use of these instruments is not really feasible with large numbers of patients, thus limiting their usefulness as a general screening instrument. Moreover, there is a need to promote healthy aging policies in order to ensure the sustainability of existing universal public health systems and to enable their implementation in countries that currently lack them (*García de Lorenzo, Álvarez & De Man, 2012*). The latter requirement is supported by the PAHO for the elderly. Hence, it is necessary to design and validate a new instrument to evaluate malnutrition in elderly individuals. This instrument should be highly sensitive and specific and be quick to administer, in order to help reduce waiting times at primary care centers. The present study presents and validates a very simple

instrument consisting of just six questions that can be easily answered by elderly patients in a very short time. In addition, the results can be evaluated by primary care personnel (i.e., nurses, physicians and dietitians) very rapidly.

The first question in the proposed model aims to determine the patient's level of physical activity. A sedentary lifestyle has been described as an important risk factor for malnutrition (*Olivares et al., 2011*), and daily physical activity can be a protective factor (*Campos et al., 2003*). The current study also identified oral health as a risk factor for malnutrition, as previously reported by others (*Porras, 2010*). Psychological problems, such as depression or dementia, was the most important factor in our scoring system (by giving the highest score = 4) to calculate the risk for the outcome. This factor has also been described as a risk factor for malnutrition (*Lee et al., 2006*; *Mamhidir et al., 2006*; *Gutiérrez et al., 2011*). General disease status, as measured by diabetes and hypertension, was confirmed to be a malnutrition risk factor, as has been previously suggested (*Jürschik et al., 2008*; *Vischer et al., 2010*; *Sanz París et al., 2013*). Finally, in this new version of the scoring system, living with family was a protective factor for our outcome. We consider this result to be logical, as the family pays more attention to the nutritional status of the elderly (*Aliabadi et al., 2008*).

### Implications for research and/or practice

These six questions and the scoring system shown in Fig. 1 constitute a new scale of malnutrition risk. According to this scale, individuals who obtain fewer than zero points have a low risk (first quartile) and do not require additional screening. Those who obtain zero points have a moderate risk (second quartile) and should proceed to the first 6 questions of the MNA test (short-form version). Those who obtain between one and three points are at high risk (third quartile) and should proceed to the complete MNA test. Finally, individuals with scores over four have a very high risk (forth quartile) and should be considered to be malnourished. Application of this instrument now requires validation mainly in the target population, in other geographical areas and in other populations (men, diabetic and hypertensive patients, and taking into account the physical activity status . . . ).

## CONCLUSIONS

A screening test for malnutrition in elderly women is presented. The test is based on six very simple questions regarding physical activity, living with family, hypertension, diabetes, dentition and psychological problems. Because these questions are very quick and easy to evaluate, they can replace more complex tests, such as the MNA, as a preferred diagnostic confirmation instrument. However, the test should be used with caution until additional validation studies in other geographical areas have been performed to support the present findings. These validation studies should be performed in the target population (elderly women), as well as in other populations (men, diabetic and hypertensive patients, and taking into account the physical activity status . . . ).

## ACKNOWLEDGEMENT

We wish to thank Ian Johnstone for help with the English language version of the text.

### Funding

The authors received no funding for this work.

### Competing Interests

Antonio Palazon-Bru serves as an Academic Editor for PeerJ.

### Author Contributions

- Tranquilina Gutiérrez-Gómez and Isabel Peñarrieta-de Córdova conceived and designed the experiments, performed the experiments, reviewed drafts of the paper.
- Ernesto Cortés conceived and designed the experiments, wrote the paper, reviewed drafts of the paper.
- Antonio Palazón-Bru conceived and designed the experiments, analyzed the data, wrote the paper, prepared figures and/or tables, reviewed drafts of the paper.
- Vicente Francisco Gil-Guillén and Rosa María Ferrer-Diego conceived and designed the experiments, reviewed drafts of the paper.

### Human Ethics

The following information was supplied relating to ethical approvals (i.e., approving body and any reference numbers):

The Autonomous University of Tamaulipas and the ISSSTE approved the study (code: 08-06-01), and all the women provided written informed consent. The study complied with the provisions of the Mexican General Health Law.

### Supplemental Information

Supplemental information for this article can be found online at http://dx.doi.org/10.7717/peerj.1316#supplemental-information.

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
