# Peer review of "Six simple questions to detect malnutrition or malnutrition risk in elderly women"

_PeerJ, doi:10.7717/peerj.1316_

## Round 0.1 · original submission · Major Revisions

· Academic Editor

Major Revisions

I think this manuscript deals with an interesting topic and it provides a potentially interesting new tool, tested in a population of a reasonable size.

Nevertheless, some major modifications should be carried out in order to make it acceptable for publication. Besides the different aspects raised by the reviewers, I think it is especially important that the new test applied by the authors is properly explained in the Materials and Methods section. Also, it is not completely clear how the observed and the expected outcomes shown in Figure 3 are calculated. Finally, there should be a deeper discussion comparing the advantages of this new test as compared with MNA-short, which is a validated procedure equally with six questions.

·

Basic reporting

I find it very interesting to study but I would need more information

Experimental design

You should have done more nutritional assessment test to compare with the one you propose

Validity of the findings

I think they have done the full MNA .
You should compare it in front of your test , using as gold stantandar the MNA.
I think that it might be interesting to compare the results of your nutritional assessment among different groups : diabetic and non-diabetic , and the same in hypertensive or socioeconomic status ...
You have data on clinical outcomes such as falls in the last year. This is a fact to be used for the usefulness of your test.

Additional comments

I think that you need a gold standard to compare with the results of your test.

·

Basic reporting

No Comments

Experimental design

Methods are not clear enough and they are confusing in some places.

Validity of the findings

Some statistical outcomes should be added.
The conclusion and aim of the study should be more consistent.

Additional comments

I have reviewed the manuscript entitled Six simple questions to detect malnutrition or malnutrition risk in elderly women with 2015:06:5443:0:0:REVIEW ID.
My comments are the following;
1. The title and issue are exciting and interesting. I aggree with the authors that MNA full form is not practical enough in some of outpatient clinics however, MNA short form should not be ignored.
2. The method for evaluating the relevant questions and comparing the new scale with MNA is correct however the values of sensitivity and specificity should be definitely given.
3. The target population is female elderly. Because this new scale was conducted on elderly women, it should be concluded to perform the future validity studies on women?
4. Abstract is well written and the recommendations are well designed.
5. The language is fluent and understandable.
6. Although NUTRITION is main factor in malNUTRITION, the researchers did not take nutritional intake as a risk factor? Please discuss and explain with references!…
7. In materials and method section, there are a few subjects that are not really comprehensible;
7a. The women were divided into 2 groups based on physical activity, however no further process goes on with these two groups.
7b. The sample were fasting for 6 hours prior to the data collection day. There is no relevant or related finding. Did the researchers do any biochemical analysis? If no why were the women fasting?
7c. “For every 25 outcomes” is not clear enough. Please explain in details. And what does 9,907 mean?
8. The new scale is recommended for nurses and physicians to practice, what about dietitians? Dietitians are the first health care professionals who will evaluate patient’s/subject’s nutritional status!
9. The psychological problems lead to the highest rate for malnutrition in this scale. However, it was not introduced and discussed that it is the most important (by giving the highest score=4) risk factor for malnutrition. Please give a convincing information with references.
10. On the other hand, living with family becomes one of the most important risk factors but there is not a reference fort his risk factor except fort he authors’ own thoughts.
11. On page 9, Line 171; “these six questions …” are not belong to Figure 1, it should be referred to Figure 2.
12. Quartilles are not correct in terms of statistics. There should be 3 quartilles. Please consult with a biostatistician.
13. The aim and the conclusion are not consistent. The purpose of the study should be better explained. For example; “To develop a model to be used as a pre-screening tool for malnutrition”.
14. The first future study should be conducted on the target population.

---

## Round 0.2 · accepted · Accept

· Academic Editor

Accept

I think the manuscript has improved with the modifications included by the authors and it is now acceptable for publication.